# DensePure: Understanding Diffusion Models towards Adversarial Robustness

## Abstract

Diffusion models have been recently employed to improve certified robustness through the process of denoising. However, the theoretical understanding of why diffusion models are able to improve the certified robustness is still lacking, preventing from further improvement. In this study, we close this gap by analyzing the fundamental properties of diffusion models and establishing the conditions under which they can enhance certified robustness. This deeper understanding allows us to propose a new method DensePure, designed to improve the certified robustness of a pretrained model (i.e. classifier). Given an (adversarial) input, DensePure consists of multiple runs of denoising via the reverse process of the diffusion model (with different random seeds) to get multiple reversed samples, which are then passed through the classifier, followed by majority voting of inferred labels to make the final prediction. This design of using multiple runs of denoising is informed by our theoretical analysis of the conditional distribution of the reversed sample. Specifically, when the *data* density of a clean sample is high, its conditional density under the reverse process in a diffusion model is also high; thus sampling from the latter conditional distribution can purify the adversarial example and return the corresponding clean sample with a high probability. By using the highest density point in the conditional distribution as the reversed sample, we identify the robust region of a given instance under the diffusion model's reverse process. We show that this robust region is a union of multiple convex sets, and is potentially much larger than the robust regions identified in previous works. In practice, DensePure can approximate the label of the high density region in the conditional distribution so that it can enhance certified robustness. We conduct extensive experiments to demonstrate the effectiveness of DensePure by evaluating its certified robustness given a standard model via randomized smoothing. We show that DensePure is consistently better than existing methods on ImageNet, with 7% improvement on average.

## 1 Introduction

Diffusion models have been shown to be a powerful image generation tool (Ho et al., 2020; Song et al., 2021b) owing to their iterative diffusion and denoising processes. These models have achieved state-of-the-art performance on sample quality (Dhariwal & Nichol, 2021; Vahdat et al., 2021) as well as effective mode coverage (Song et al., 2021a). A diffusion model usually consists of two processes: (i) a forward diffusion process that converts data to noise by gradually adding noise to the input, and (ii) a reverse generative process that starts from noise and generates data by denoising one step at a time (Song et al., 2021b).

Given the natural denoising property of diffusion models, *empirical* studies have leveraged them to perform adversarial purification (Nie et al., 2022; Wu et al., 2022; Carlini et al., 2022). For instance,

Nie et al. (2022) introduce a diffusion model based purification model *DiffPure*. They *empirically* show that by carefully choosing the amount of Gaussian noises added during the diffusion process, adversarial perturbations can be removed while preserving the true label semantics. Despite the significant empirical results, there is no provable guarantee of the achieved robustness. Carlini et al. (2022) instantiate the randomized smoothing approach with the diffusion model to offer a *provable guarantee* of model robustness against $L_2$-norm bounded adversarial example. However, they do not provide a theoretical understanding of why and how the diffusion models contribute to such nontrivial certified robustness.

**Our Approach.** We theoretically analyze the fundamental properties of diffusion models to understand why and how it enhances certified robustness. This deeper understanding allows us to propose a new method DensePure to improve the certified robustness of any given classifier by more effectively using the diffusion model. It consists of a pretrained diffusion model and a pretrained classifier. DensePure incorporates two steps: (i) using the reverse process of the diffusion model to obtain a sample of the posterior data distribution conditioned on the adversarial input; and (ii) repeating the reverse process multiple times with different random seeds to approximate the label of high density region in the conditional distribution via a majority vote. In particular, given an adversarial input, we repeatedly feed it into the reverse process of the diffusion model to get multiple reversed examples and feed them into the classifier to get their labels. We then apply the *majority vote* on the set of labels to get the final predicted label.

DensePure is inspired by our theoretical analysis, where we show that the diffusion model reverse process provides a conditional distribution of the reversed sample given an adversarial input, and sampling from this conditional distribution enhances the certified robustness. Specifically, we prove that when the data density of clean samples is high, it is a sufficient condition for the conditional density of the reversed samples to be also high. Therefore, in DensePure, samples from the conditional distribution can recover the ground-truth labels with a high probability.

For the convenience of understanding and rigorous analysis, we use the highest density point in the conditional distribution as the deterministic reversed sample for the classifier prediction. We show that the robust region for a given sample under the diffusion model's reverse process is the union of multiple convex sets, each surrounding a region around the ground-truth label. Compared with the robust region of previous work (Cohen et al., 2019), which only focuses on the neighborhood of *one* region with the ground-truth label, such union of multiple convex sets has the potential to provide a much larger robust region. Moreover, the characterization implies that the size of robust regions is affected by the relative density and the distance between data regions with the ground-truth label and those with other labels.

We conduct extensive experiments on ImageNet and CIFAR-10 datasets under different settings to evaluate the certifiable robustness of DensePure. In particular, we follow the setting from Carlini et al. (2022) and rely on randomized smoothing to certify robustness to adversarial perturbations bounded in the $\mathcal{L}_2$-norm. We show that DensePure achieves the new state-of-the-art *certified robustness* on the clean model without tuning any model parameters (off-the-shelf). On ImageNet, it achieves a consistently higher certified accuracy than the existing methods among every $\sigma$ at every radius $\epsilon$, 7% improvement on average.

## 2  Preliminaries and Backgrounds

**Continuous-Time Diffusion Model.** The diffusion model has two components: the *diffusion process* followed by the *reverse process*. Given an input random variable $\mathbf{x}_0 \sim p$, the diffusion process adds isotropic Gaussian noises to the data so that the diffused random variable at time $t$ is $\mathbf{x}_t = \sqrt{\alpha_t}(\mathbf{x}_0 + \boldsymbol{\epsilon}_t)$, s.t., $\boldsymbol{\epsilon}_t \sim \mathcal{N}(\mathbf{0}, \sigma_t^2 \boldsymbol{I})$, and $\sigma_t^2 = (1 - \alpha_t)/\alpha_t$, and we denote $\mathbf{x}_t \sim p_t$. The forward diffusion process can also be defined by the stochastic differential equation

$$d\boldsymbol{x} = h(\boldsymbol{x}, t)dt + g(t)d\boldsymbol{w}, \qquad \text{(SDE)}$$

where $\boldsymbol{x}_0 \sim p$, $h : \mathbb{R}^d \times \mathbb{R} \mapsto \mathbb{R}^d$ is the drift coefficient, $g : \mathbb{R} \mapsto \mathbb{R}$ is the diffusion coefficient, and $\boldsymbol{w}(t) \in \mathbb{R}^n$ is the standard Wiener process.

Under mild conditions C.1, the reverse process exists and removes the added noise by solving the reverse-time SDE (Anderson, 1982)

$$d\hat{\boldsymbol{x}} = [h(\hat{\boldsymbol{x}}, t) - g(t)^2 \nabla_{\hat{\boldsymbol{x}}} \log p_t(\hat{\boldsymbol{x}})]dt + g(t)d\overline{\boldsymbol{w}}, \qquad \text{(reverse-SDE)}$$

89    where $dt$ is an infinitesimal reverse time step, and $\overline{w}(t)$ is a reverse-time standard Wiener process.

90    In our context, we use the conventions of VP-SDE (Song et al., 2021b) where $h(\boldsymbol{x};t) := -\frac{1}{2}\gamma(t)x$

91    and $g(t) := \sqrt{\gamma(t)}$ with $\gamma(t)$ positive and continuous over $[0,1]$, such that $x(t) = \sqrt{\alpha_t}x(0) +$

92    $\sqrt{1-\alpha_t}\epsilon$ where $\alpha_t = e^{-\int_0^t \gamma(s)ds}$ and $\epsilon \sim \mathcal{N}(\mathbf{0},\boldsymbol{I})$. We use $\{\mathbf{x}_t\}_{t\in[0,1]}$ and $\{\hat{\mathbf{x}}_t\}_{t\in[0,1]}$ to denote

93    the diffusion process and the reverse process generated by SDE and reverse-SDE respectively, which

94    follow the same distribution.

95    The formulations of Discrete-Time Diffusion Model (or DDPM (Ho et al., 2020)) and Randomized

96    Smoothing are in the appendix.

## 97   3   Theoretical Analysis

98    In this section, we theoretically analyze why and how the diffusion model can enhance the robustness

99    of a given classifier. We will analyze directly on SDE and reverse-SDE as they generate the same

100    stochastic processes $\{\mathbf{x}_t\}_{t\in[0,T]}$ and the literature works establish an approximation on reverse-

101    SDE (Song et al., 2021b; Ho et al., 2020).

102    We first show that given a diffusion model, solving reverse-SDE will generate a conditional distribu-

103    tion based on the scaled adversarial sample, which will have high density on data region with high

104    *data* density and near to the adversarial sample in Theorem 3.1. See detailed conditions in C.1.

105    **Theorem 3.1.** *Under conditions C.1, solving equation reverse-SDE starting from time $t$ and sample*

106    $\boldsymbol{x}_{a,t} = \sqrt{\alpha_t}\boldsymbol{x}_a$ *will generate a reversed random variable* $\hat{\mathbf{x}}_0$ *with density* $\mathbb{P}\left(\hat{\mathbf{x}}_0 = \boldsymbol{x} | \hat{\mathbf{x}}_t = \boldsymbol{x}_{a,t}\right) \propto$

107    $p(\boldsymbol{x}) \cdot \frac{1}{\sqrt{(2\pi\sigma_t^2)^n}} \exp\left(\frac{-||\boldsymbol{x}-\boldsymbol{x}_a||_2^2}{2\sigma_t^2}\right)$, *where $p$ is the data distribution,* $\sigma_t^2 = \frac{1-\alpha_t}{\alpha_t}$ *is the variance of*

108    *Gaussian noise added at time $t$ in the diffusion process.*

109    *Proof.* (sketch) Under conditions C.1, we know $\{\mathbf{x}_t\}_{t\in[0,1]}$ and $\{\hat{\mathbf{x}}_t\}_{t\in[0,1]}$ follow the same distri-

110    bution, and then the rest proof follows Bayes' Rule. $\qquad\square$

111    Please see the full proofs of this and the following theorems in Appendix C.3.

112    **Remark 1.** *Note that* $\mathbb{P}\left(\hat{\mathbf{x}}_0 = \boldsymbol{x} | \hat{\mathbf{x}}_t = \boldsymbol{x}_{a,t}\right) > 0$ *if and only if $p(\boldsymbol{x}) > 0$, thus the generated reverse*

113    *sample will be on the data region where we train classifiers.*

114    In Theorem 3.1, the conditional density $\mathbb{P}\left(\hat{\mathbf{x}}_0 = \boldsymbol{x} | \hat{\mathbf{x}}_t = \boldsymbol{x}_{a,t}\right)$ is high only if both $p(\boldsymbol{x})$ and the

115    Gaussian term have high values, i.e., $\boldsymbol{x}$ has high *data* density and is close to the adversarial sample

116    $\boldsymbol{x}_a$. The latter condition is reasonable since adversarial perturbations are typically bounded due to

117    budget constraints. Then, the above argument implies that a reversed sample will have the ground-

118    truth label with a high probability if data region with the ground-truth label has high enough *data*

119    density.

120    For the convenience of theoretical analysis and understanding, we take the point with high-

121    est conditional density $\mathbb{P}\left(\hat{\mathbf{x}}_0 = \boldsymbol{x} | \hat{\mathbf{x}}_t = \boldsymbol{x}_{a,t}\right)$ as the reversed sample, defined as $\mathcal{P}(\boldsymbol{x}_a;t) :=$

122    $\arg\max_{\boldsymbol{x}} \mathbb{P}\left(\hat{\mathbf{x}}_0 = \boldsymbol{x} | \hat{\mathbf{x}}_t = \boldsymbol{x}_{a,t}\right)$. $\mathcal{P}(\boldsymbol{x}_a;t)$ is a representative of the high density data region in

123    the conditional distribution and $\mathcal{P}(\cdot;t)$ is a deterministic purification model. In the following, we

124    characterize the robust region for data region with ground-truth label under $\mathbb{P}\left(\cdot;t\right)$. The robust re-

125    gion and the robust radius for a general deterministic purification model given a classifier are defined

126    below.

127    **Definition 3.2** (Robust Region and Robust Radius). *Given a classifier $f$ and a point $\boldsymbol{x}_0$, let*

128    $\mathcal{G}(\boldsymbol{x}_0) := \{\boldsymbol{x} : f(\boldsymbol{x}) = f(\boldsymbol{x}_0)\}$ *be the data region where samples have the same label as $\boldsymbol{x}_0$.*

129    *Then given a deterministic purification model $\mathcal{P}(\cdot;\psi)$ with parameter $\psi$, we define the robust re-*

130    *gion of $\mathcal{G}(\boldsymbol{x}_0)$ under $\mathcal{P}$ and $f$ as* $\mathcal{D}_{\mathcal{P}}^f\left(\mathcal{G}(\boldsymbol{x}_0);\psi\right) := \{\boldsymbol{x} : f(\mathcal{P}(\boldsymbol{x};\psi)) = f(\boldsymbol{x}_0)\}$, *i.e., the set of $\boldsymbol{x}$*

131    *such that purified sample $\mathcal{P}(\boldsymbol{x};\psi)$ has the same label as $\boldsymbol{x}_0$ under $f$. Further, we define the robust*

132    *radius of $\boldsymbol{x}_0$ as* $r_{\mathcal{P}}^f(\boldsymbol{x}_0;\psi) := \max\left\{r : \boldsymbol{x}_0 + ru \in \mathcal{D}_{\mathcal{P}}^f\left(\boldsymbol{x}_0;\psi\right) , \forall||u||_2 \le 1\right\}$, *i.e., the radius of*

133    *maximum inclined ball of $\mathcal{D}_{\mathcal{P}}^f\left(\boldsymbol{x}_0;\psi\right)$ centered around $\boldsymbol{x}_0$. We will omit $\mathcal{P}$ and $f$ when it is clear*

134    *from the context and write $\mathcal{D}\left(\mathcal{G}(\boldsymbol{x}_0);\psi\right)$ and $r(\boldsymbol{x}_0;\psi)$ instead.*

135    **Remark 2.** *In Definition 3.2, the robust region (resp. radius) is defined for each class (resp. point).*

136    *When using the point with highest $\mathbb{P}\left(\hat{\mathbf{x}}_0 = \boldsymbol{x} | \hat{\mathbf{x}}_t = \boldsymbol{x}_{a,t}\right)$ as the reversed sample, $\psi := t$.*

137 Now given a sample $\boldsymbol{x}_0$ with ground-truth label, we are ready to characterize the robust region
138 $\mathcal{D}\left(\mathcal{G}(\boldsymbol{x}_0);\psi\right)$ under purification model $\mathcal{P}(\cdot;t)$ and classifier $f$. Intuitively, if the adversarial sample
139 $\boldsymbol{x}_a$ is near to $\boldsymbol{x}_0$ (in Euclidean distance), $\boldsymbol{x}_a$ keeps the same label semantics of $\boldsymbol{x}_0$ and so as the
140 purified sample $\mathcal{P}(\boldsymbol{x}_a;t)$, which implies that $f\left(\mathcal{P}(\boldsymbol{x}_a;\psi)\right) = f(\boldsymbol{x}_0)$. However, the condition that
141 $\boldsymbol{x}_a$ is near to $\boldsymbol{x}_0$ is sufficient but not necessary since we can still achieve $f\left(\mathcal{P}(\boldsymbol{x}_a;\psi)\right) = f(\boldsymbol{x}_0)$
142 if $\boldsymbol{x}_a$ is near to any sample $\tilde{\boldsymbol{x}}_0$ with $f\left(\mathcal{P}(\tilde{\boldsymbol{x}}_0;\psi)\right) = f(\boldsymbol{x}_0)$. In the following, we will show that
143 the robust region $\mathcal{D}\left(\mathcal{G}(\boldsymbol{x}_0);\psi\right)$ is the union of the convex robust sub-regions surrounding every $\tilde{\boldsymbol{x}}_0$
144 with the same label as $\boldsymbol{x}_0$. The following theorem characterizes the convex robust sub-region and
145 robust region respectively.

**Theorem 3.3.** *Under conditions C.1 and classifier $f$, let $\boldsymbol{x}_0$ be the sample with ground-truth label*
147 *and $\boldsymbol{x}_a$ be the adversarial sample, then (i) the purified sample $\mathcal{P}(\boldsymbol{x}_a;t)$ will have the ground-truth*
148 *label if $\boldsymbol{x}_a$ falls into the following convex set,*

$$\mathcal{D}_{sub}\left(\boldsymbol{x}_0;t\right) := \bigcap_{\left\{\boldsymbol{x}_0':f(\boldsymbol{x}_0')\neq f(\boldsymbol{x}_0)\right\}} \left\{\boldsymbol{x}_a : (\boldsymbol{x}_a-\boldsymbol{x}_0)^\top(\boldsymbol{x}_0'-\boldsymbol{x}_0) < \sigma_t^2 \log\left(\frac{p(\boldsymbol{x}_0)}{p(\boldsymbol{x}_0')}\right) + \frac{||\boldsymbol{x}_0'-\boldsymbol{x}_0||_2^2}{2}\right\},$$

149 *and further, (ii) the purified sample $\mathcal{P}(\boldsymbol{x}_a;t)$ will have the ground-truth label if and only if $\boldsymbol{x}_a$ falls*
150 *into the following set, $\mathcal{D}\left(\mathcal{G}(\boldsymbol{x}_0);t\right) := \bigcup_{\tilde{\boldsymbol{x}}_0:f(\tilde{\boldsymbol{x}}_0)=f(\boldsymbol{x}_0)} \mathcal{D}_{sub}\left(\tilde{\boldsymbol{x}}_0;t\right)$. In other words, $\mathcal{D}\left(\mathcal{G}(\boldsymbol{x}_0);t\right)$*
151 *is the robust region for data region $\mathcal{G}(\boldsymbol{x}_0)$ under $\mathcal{P}(\cdot;t)$ and $f$.*

*Proof.* (sketch) (i). Each convex half-space defined by the inequality corresponds to a $\boldsymbol{x}_0'$ such that
153 $f(\boldsymbol{x}_0') \neq f(\boldsymbol{x}_0)$ where $\boldsymbol{x}_a$ within satisfies $\mathbb{P}(\hat{\mathbf{x}}_0=\boldsymbol{x}_0|\hat{\mathbf{x}}_t=\boldsymbol{x}_{a,t}) > \mathbb{P}(\hat{\mathbf{x}}_0=\boldsymbol{x}_0' \mid \hat{\mathbf{x}}_t=\boldsymbol{x}_{a,t})$. This
154 implies that $\mathcal{P}(\boldsymbol{x}_a;t) \neq \boldsymbol{x}_0'$ and $f\left(\mathcal{P}(\boldsymbol{x}_a;\psi)\right) = f(\boldsymbol{x}_0)$. The convexity is due to that the intersection
155 of convex sets is convex. (ii). The "if" follows directly from (i). The "only if" holds because
156 if $\boldsymbol{x}_a \notin \mathcal{D}\left(\mathcal{G}(\boldsymbol{x}_0);t\right)$, then exists $\tilde{\boldsymbol{x}}_1$ such that $f(\tilde{\boldsymbol{x}}_1) \neq f(\boldsymbol{x}_0)$ and $\mathbb{P}(\hat{\mathbf{x}}_0=\tilde{\boldsymbol{x}}_1|\hat{\mathbf{x}}_t=\boldsymbol{x}_{a,t}) >$
157 $\mathbb{P}(\hat{\mathbf{x}}_0=\tilde{\boldsymbol{x}}_0|\hat{\mathbf{x}}_t=\boldsymbol{x}_{a,t}), \forall \tilde{\boldsymbol{x}}_0$ s.t. $f(\tilde{\boldsymbol{x}}_0)=f(\boldsymbol{x}_0)$, and thus $f\left(\mathcal{P}(\boldsymbol{x}_a;\psi)\right) \neq f(\boldsymbol{x}_0)$. $\qquad\square$

**Remark 3.** *Theorem 3.3 implies that when data region $\mathcal{G}(\boldsymbol{x}_0)$ has higher data density and larger*
159 *distances to data regions with other labels, it tends to have larger robust region and points in data*
160 *region tends to have larger radius.*

161 In the literature, people focus more on the robust radius (lower bound) $r\left(\mathcal{G}(\boldsymbol{x}_0);t\right)$ (Cohen et al.,
162 2019; Carlini et al., 2022), which can be obtained by finding the maximum inclined ball inside
163 $\mathcal{D}\left(\mathcal{G}(\boldsymbol{x}_0);t\right)$ centering $\boldsymbol{x}_0$. Note that although $\mathcal{D}_{\text{sub}}\left(\boldsymbol{x}_0;t\right)$ is convex, $\mathcal{D}\left(\mathcal{G}(\boldsymbol{x}_0);t\right)$ is generally
164 not. Therefore, finding $r\left(\mathcal{G}(\boldsymbol{x}_0);t\right)$ is a non-convex optimization problem. In particular, it can be
165 formulated into a disjunctive optimization problem with integer indicator variables, which is typi-
166 cally NP-hard to solve. One alternative could be finding the maximum inclined ball in $\mathcal{D}_{\text{sub}}\left(\boldsymbol{x}_0;t\right)$,
167 which can be formulated into a convex optimization problem whose optimal value provides a lower
168 bound for $r\left(\mathcal{G}(\boldsymbol{x}_0);t\right)$. However, $\mathcal{D}\left(\mathcal{G}(\boldsymbol{x}_0);t\right)$ has the potential to provide much larger robustness
169 radius because it might connect different convex robust sub-regions into one.

170 In practice, we cannot guarantee to establish an exact reverse process like reverse-SDE but instead
171 try to establish an approximate reverse process to mimic the exact one. As long as the approximate
172 reverse process is close enough to the exact reverse process, they will generate close enough con-
173 ditional distributions based on the adversarial sample. Then the density and locations of the data
174 regions in two conditional distributions will not differ much and so is the robust region for each
175 data region. We take the score-based diffusion model in Song et al. (2021b) for an example and
176 demonstrate Theorem 3.4 to bound the KL-divergnece between conditional distributions generated
177 by reverse-SDE and score-based diffusion model. Ho et al. (2020) showed that using variational
178 inference to fit DDPM is equivalent to optimizing an objective resembling score-based diffusion
179 model with a specific weighting scheme, so the results can be extended to DDPM.

**Theorem 3.4.** *Under score-based diffusion model Song et al. (2021b) and conditions C.1, we have*
181 $D_{KL}(\mathbb{P}(\hat{\mathbf{x}}_0=\boldsymbol{x} \mid \hat{\mathbf{x}}_t=\boldsymbol{x}_{a,t})\|\mathbb{P}(\mathbf{x}_0^\theta=\boldsymbol{x} \mid \mathbf{x}_t^\theta=\boldsymbol{x}_{a,t})) = \mathcal{J}_{\text{SM}}(\theta,t;\lambda(\cdot))$, *where* $\{\hat{\boldsymbol{x}}_\tau\}_{\tau\in[0,t]}$ *and*
182 $\{\boldsymbol{x}_\tau^\theta\}_{\tau\in[0,t]}$ *are stochastic processes generated by reverse-SDE and score-based diffusion model*
183 *respectively,* $\mathcal{J}_{\text{SM}}(\theta,t;\lambda(\cdot)) := \frac{1}{2}\int_0^t \mathbb{E}_{p_\tau(\mathbf{x})}\left[\lambda(\tau)\left\|\nabla_{\mathbf{x}}\log p_\tau(\mathbf{x}) - \boldsymbol{s}_\theta(\mathbf{x},\tau)\right\|_2^2\right]\mathrm{d}\tau$, $\boldsymbol{s}_\theta(\mathbf{x},\tau)$ *is the*
184 *score function to approximate* $\nabla_{\mathbf{x}}\log p_\tau(\mathbf{x})$, *and* $\lambda : \mathbb{R} \to \mathbb{R}$ *is any weighting scheme used in the*
185 *training score-based diffusion models.*

| | | CIFAR-10 | | | | Certified Accuracy at $\epsilon$(%) ImageNet | | | | |
|---|---|---|---|---|---|---|---|---|---|---|
| Method | Off-the-shelf | 0.25 | 0.5 | 0.75 | 1.0 | 0.5 | 1.0 | 1.5 | 2.0 | 3.0 |
| PixelDP (Lecuyer et al., 2019) | ✗ | (71.0)22.0 | (44.0)2.0 | - | - | (33.0)16.0 | - | - | - | - |
| RS (Cohen et al., 2019) | ✗ | (75.0)61.0 | (75.0)43.0 | (65.0)32.0 | (65.0)23.0 | (67.0)49.0 | (57.0)37.0 | (57.0)29.0 | (44.0)19.0 | (44.0)12.0 |
| SmoothAdv (Salman et al., 2019a) | ✗ | (82.0)68.0 | (76.0)54.0 | (68.0)41.0 | (64.0)32.0 | (63.0)54.0 | (56.0)42.0 | (56.0)34.0 | (41.0)26.0 | (41.0)18.0 |
| Consistency (Jeong & Shin, 2020) | ✗ | (77.8)68.8 | (75.8)58.1 | (72.9)48.5 | (52.3)37.8 | (55.0)50.0 | (55.0)44.0 | (55.0)34.0 | (41.0)24.0 | (41.0)17.0 |
| MACER (Zhai et al., 2020) | ✗ | (81.0)71.0 | (81.0)59.0 | (66.0)46.0 | (66.0)38.0 | (68.0)57.0 | (64.0)43.0 | (64.0)31.0 | (48.0)25.0 | (48.0)14.0 |
| Boosting (Horváth et al., 2021) | ✗ | (83.4)70.6 | (76.8)60.4 | (71.6)**52.4** | (73.0)**38.8** | (65.6)57.0 | (57.0)44.6 | (57.0)38.4 | (44.6)28.6 | (38.6)21.2 |
| SmoothMix (Jeong et al., 2021) | ✓ | (77.1)67.9 | (77.1)57.9 | (74.2)47.7 | (61.8)37.2 | (55.0)50.0 | (55.0)43.0 | (55.0)38.0 | (40.0)26.0 | (40.0)17.0 |
| Denoised (Salman et al., 2020) | ✓ | (72.0)56.0 | (62.0)41.0 | (62.0)28.0 | (44.0)19.0 | (60.0)33.0 | (38.0)14.0 | (38.0)6.0 | - | - |
| Lee (Lee, 2021) | ✓ | 60.0 | 42.0 | 28.0 | 19.0 | 41.0 | 24.0 | 11.0 | - | - |
| Carlini (Carlini et al., 2022) | ✓ | (88.0)73.8 | (88.0)56.2 | (88.0)41.6 | (74.2)31.0 | (77.0)71.0 | (74.0)54.0 | (74.0)46.0 | (59.0)29.0 | (59.0)22.0 |
| **Ours** | ✓ | (87.6)**76.6** | (87.6)**64.6** | (87.6)50.4 | (73.6)37.4 | (80.0)**76.0** | (75.0)**62.0** | (75.0)**49.0** | (61.0)**37.0** | (61.0)**26.0** |

Table 1: Certified accuracy compared with existing works. The certified accuracy at $\epsilon = 0$ for each model is in the parentheses. The certified accuracy for each cell is from the respective papers except Carlini et al. (2022). Our diffusion model and classifier are the same as Carlini et al. (2022), where the off-the-shelf classifier uses ViT-based architectures trained on a large dataset (ImageNet-22k).

*Proof.* (sketch) Let $\boldsymbol{\mu}_t$ and $\boldsymbol{\nu}_t$ be the path measure for reverse processes $\{\hat{\mathbf{x}}_\tau\}_{\tau \in [0,t]}$ and $\{\mathbf{x}_\tau^\theta\}_{\tau \in [0,t]}$ respectively based on the $\boldsymbol{x}_{a,t}$. Under conditions C.1, $\boldsymbol{\mu}_t$ and $\boldsymbol{\nu}_t$ are uniquely defined and the KL-divergence can be computed via the Girsanov theorem Oksendal (2013). □

**Remark 4.** *Theorem 3.4 shows that if the training loss is smaller, the conditional distributions generated by reverse-SDE and score-based diffusion model are closer, and are the same if the training loss is zero.*

## 4 DensePure

Inspired by the theoretical analysis, we introduce DensePure and show how to calculate its certified robustness radius via the randomized smoothing algorithm.

**Framework.** Our framework, DensePure, consists of two components: (1) an off-the-shelf diffusion model with reverse process $\mathbf{rev}$ and (2) an off-the-shelf base classifier $f$.

Given an input $\boldsymbol{x}$, we feed it into the reverse process $\mathbf{rev}$ of the diffusion model to get the reversed sample $\mathbf{rev}(\boldsymbol{x})$ and then repeat the above process $K$ times to get $K$ reversed samples $\{\mathbf{rev}(\boldsymbol{x})_1, \cdots, \mathbf{rev}(\boldsymbol{x})_K\}$. We feed the above $K$ reversed samples into the classifier to get the corresponding prediction $\{f(\mathbf{rev}(\boldsymbol{x})_1), \cdots, f(\mathbf{rev}(\boldsymbol{x})_K)\}$ and then apply the *majority vote*, termed **MV**, on these predictions to get the final predicted label $\hat{y} = \mathbf{MV}(\{f(\mathbf{rev}(\boldsymbol{x})_1), \cdots, f(\mathbf{rev}(\boldsymbol{x})_K)\}) = \arg\max_c \sum_{i=1}^K \mathbf{1}\{f(\mathbf{rev}(\boldsymbol{x})_i) = c\}$ .

**Certified Robustness of DensePure with Randomized Smoothing.**

We show how DensePure can calculate certified robustness of DensePure via RS, which offers robustness guarantees for a model under a $L_2$-norm ball. In particular, we follow the similar setting of Carlini et al. (2022) which uses a DDPM-based diffusion model. The details are in the appendix.

## 5 Experiments

In this section, we use DensePure to evaluate certified robustness on two standard datasets, CIFAR-10 (Krizhevsky et al., 2009) and ImageNet (Deng et al., 2009).

**Experimental settings** We follow the experimental setting from Carlini et al. (2022). Specifically, for CIFAR-10, we use the 50-M unconditional improved diffusion model from Nichol & Dhariwal (2021) as the diffusion model. We select ViT-B/16 model Dosovitskiy et al. (2020) pretrained on ImageNet-21k and finetuned on CIFAR-10 as the classifier, which could achieve 97.9% accuracy on CIFAR-10. For ImageNet, we use the unconditional 256×256 guided diffusion model from Dhariwal & Nichol (2021) as the diffusion model and pretrained BEiT large model (Bao et al., 2021) trained on ImageNet-21k as the classifier, which could achieve 88.6% top-1 accuracy on validation set of ImageNet-1k. We select three different noise levels $\sigma \in \{0.25, 0.5, 1.0\}$ for certification. For the parameters of DensePure , we set $K = 40$ and $b = 10$ except the results in ablation study. The details about the baselines are in the appendix.

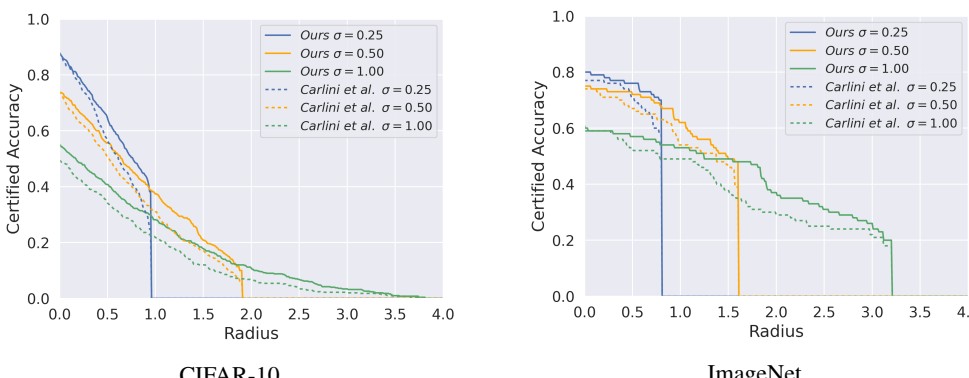

CIFAR-10           ImageNet

Figure 1: Comparing our method vs Carlini et al. (2022) on CIFAR-10 and ImageNet. The lines represent the certified accuracy with different $L_2$ perturbation bound with different Gaussian noise $\sigma \in \{0.25, 0.50, 1.00\}$.

**Main Results** We compare our results with other baselines. The results are shown in Table 1.

For CIFAR-10, comparing with the models which are *carefully* trained with randomized smoothing techniques in an end-to-end manner (i.e., w/o off-the-shelf classifier), we observe that our method with the standard off-the-shelf classifier outperforms them at smaller $\epsilon = \{0.25, 0.5\}$ on both CIFAR-10 and ImageNet datasets while achieves comparable performance at larger $\epsilon = \{0.75, 1.0\}$. Comparing with the non-diffusion model based methods with off-the-shelf classifier (i.e., De-noised (Salman et al., 2020) and Lee (Lee, 2021)), both our method and Carlini et al. (2022) are significantly better than them. These results verify the non-trivial adversarial robustness improvements introduced from the diffusion model. For ImageNet, our method is consistently better than all priors with a large margin.

Since both Carlini et al. (2022) and DensePure use the diffusion model, to better understand the importance of our design, that approximates the label of the high density region in the conditional distribution, we compare DensePure with Carlini et al. (2022) in a more fine-grained manner.

We show detailed certified robustness of the model among different $\sigma$ at different radius for CIFAR-10 in Figure 1-left and for ImageNet in Figure 1-right. We also present our results of certified accuracy at different $\epsilon$ in Appendix E.3. From these results, we find that our method is still consistently better at most $\epsilon$ (except $\epsilon = 0$) among different $\sigma$. The performance margin between ours and Carlini et al. (2022) will become even larger with a large $\epsilon$. These results further indicate that although the diffusion model improves model robustness, leveraging the posterior data distribution conditioned on the input instance (like DensePure) via reverse process instead of using single sample ((Carlini et al., 2022)) is the key for better robustness. Additionally, we use the off-the-shelf classifiers, which are the VIT-based architectures trained a larger dataset. In the later ablation study section, we select the CNN-based architecture wide-ResNet trained on standard dataset from scratch. Our method still achieves non-trivial robustness.

## 6 Conclusion

In this work, we theoretically prove that the diffusion model could purify adversarial examples back to the corresponding clean sample with high probability, as long as the data density of the corresponding clean samples is high enough. Our theoretical analysis characterizes the conditional distribution of the reversed samples given the adversarial input, generated by the diffusion model reverse process. Using the highest density point in the conditional distribution as the deterministic reversed sample, we identify the robust region of a given instance under the diffusion model reverse process, which is potentially much larger than previous methods. Our analysis inspires us to propose an effective pipeline DensePure, for adversarial robustness. We conduct comprehensive experiments to show the effectiveness of DensePure by evaluating the certified robustness via the randomized smoothing algorithm. Note that DensePure is an off-the-shelf pipeline that does not require training a smooth classifier. Our results show that DensePure achieves the new SOTA certified robustness for perturbation with $\mathcal{L}_2$-norm. We hope that our work sheds light on an in-depth understanding of the diffusion model for adversarial robustness.

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

 # Appendix

 Here is the appendix.

## A  Notations

| | |
|---|---|
| $p$ | data distribution |
| $\mathbb{P}(A)$ | probability of event $A$ |
| $\mathcal{C}^k$ | set of functions with continuous $k$-th derivatives |
| $\boldsymbol{w}(t)$ | standard Wiener Process |
| $\overline{\boldsymbol{w}}(t)$ | reverse-time standard Wiener Process |
| $h(\boldsymbol{x}, t)$ | drift coefficient in SDE |
| $g(t)$ | diffusion coefficient in SDE |
| $\alpha_t$ | scaling coefficient at time $t$ |
| $\sigma_t^2$ | variance of added Gaussian noise at time $t$ |
| $\{\mathbf{x}_t\}_{t\in[0,1]}$ | diffusion process generated by SDE |
| $\{\hat{\mathbf{x}}_t\}_{t\in[0,1]}$ | reverse process generated by reverse-SDE |
| $p_t$ | distribution of $\mathbf{x}_t$ and $\hat{\mathbf{x}}_t$ |
| $\{\mathbf{x}_1, \mathbf{x}_2, \ldots, \mathbf{x}_N\}$ | diffusion process generated by DDPM |
| $\{\beta_i\}_{i=1}^N$ | pre-defined noise scales in DDPM |
| $\boldsymbol{\epsilon}_a$ | adversarial attack |
| $\boldsymbol{x}_a$ | adversarial sample |
| $\boldsymbol{x}_{a,t}$ | scaled adversarial sample |
| $f(\cdot)$ | classifier |
| $g(\cdot)$ | smoothed classifier |
| $\mathbb{P}\left(\hat{\mathbf{x}}_0 = \boldsymbol{x} \mid \hat{\mathbf{x}}_t = \boldsymbol{x}_{a,t}\right)$ | density of conditional distribution generated by reverse-SDE based on $\boldsymbol{x}_{a,t}$ |
| $\mathcal{P}(\boldsymbol{x}_a; t)$ | purification model with highest density point |
| $\mathcal{G}(\boldsymbol{x}_0)$ | data region with the same label as $\boldsymbol{x}_0$ |
| $\mathcal{D}_{\mathcal{P}}^f(\mathcal{G}(\boldsymbol{x}_0); t)$ | robust region for $\mathcal{G}(\boldsymbol{x}_0)$ associated with base classifier $f$ and purification model $\mathcal{P}$ |
| $r_{\mathcal{P}}^f(\boldsymbol{x}_0; t)$ | robust radius for the point associated with base classifier $f$ and purification model $\mathcal{P}$ |
| $\mathcal{D}_{sub}(\boldsymbol{x}_0; t)$ | convex robust sub-region |
| $\boldsymbol{s}_\theta(\boldsymbol{x}, t)$ | score function |
| $\{\mathbf{x}_t^\theta\}_{t\in[0,1]}$ | reverse process generated by score-based diffusion model |
| $\mathbb{P}\left(\mathbf{x}_0^\theta = \boldsymbol{x} \mid \mathbf{x}_t^\theta = \boldsymbol{x}_{a,t}\right)$ | density of conditional distribution generated by score-based diffusion model based on $\boldsymbol{x}_{a,t}$ |
| $\lambda(\tau)$ | weighting scheme of training loss for score-based diffusion model |
| $\mathcal{J}_{\mathrm{SM}}(\theta, t; \lambda(\cdot))$ | truncated training loss for score-based diffusion model |
| $\boldsymbol{\mu}_t, \boldsymbol{\nu}_t$ | path measure for $\{\hat{\mathbf{x}}_\tau\}_{\tau\in[0,t]}$ and $\{\mathbf{x}_\tau^\theta\}_{\tau\in[0,t]}$ respectively |

# B  Related Work

Using an off-the-shelf generative model to purify adversarial perturbations has become an important direction in adversarial defense. Previous works have developed various purification methods based on different generative models, such as GANs (Samangouei et al., 2018), autoregressive generative models (Song et al., 2018), and energy-based models (Du & Mordatch, 2019; Grathwohl et al., 2020; Hill et al., 2021). More recently, as diffusion models (or score-based models) achieve better generation quality than other generative models (Ho et al., 2020; Dhariwal & Nichol, 2021), many works consider using diffusion models for adversarial purification (Nie et al., 2022; Wu et al., 2022; Sun et al., 2022) Although they have found good empirical results in defending against existing adversarial attacks (Nie et al., 2022), there is no provable guarantee about the robustness about such methods. On the other hand, certified defenses provide guarantees of robustness (Mirman et al., 2018; Cohen et al., 2019; Lecuyer et al., 2019; Salman et al., 2020; Horváth et al., 2021; Zhang et al., 2018; Raghunathan et al., 2018a,b; Salman et al., 2019b; Wang et al., 2021). They provide a lower bounder of model accuracy under constrained perturbations. Among them, approaches Lecuyer et al. (2019); Cohen et al. (2019); Salman et al. (2019a); Jeong & Shin (2020); Zhai et al. (2020); Horváth et al. (2021); Jeong et al. (2021); Salman et al. (2020); Lee (2021); Carlini et al. (2022) based on randomized smoothing (Cohen et al., 2019) show the great scalability and achieve promising performance on large network and dataset. The most similar work to us is Carlini et al. (2022), which uses diffusion models combined with standard classifiers for certified defense. They view diffusion model as blackbox without having a theoretical under- standing of why and how the diffusion models contribute to such nontrivial certified robustness.

# C  More details about Theoretical analysis

## C.1  Assumptions

    (i) The data distribution $p \in \mathcal{C}^2$ and $\mathbb{E}_{\boldsymbol{x} \sim p}[||\boldsymbol{x}||_2^2] < \infty$.

    (ii) $\forall t \in [0, T] : h(\cdot, t) \in \mathcal{C}^1, \exists C > 0, \forall \boldsymbol{x} \in \mathbb{R}^n, t \in [0, T] : ||h(\boldsymbol{x}, t)||_2 \leqslant C(1 + ||\boldsymbol{x}||_2)$.

    (iii) $\exists C > 0, \forall \boldsymbol{x}, \boldsymbol{y} \in \mathbb{R}^n : ||h(\boldsymbol{x}, t) - h(\boldsymbol{y}, t)||_2 \leqslant C||\boldsymbol{x} - \boldsymbol{y}||_2$.

    (iv) $g \in \mathcal{C}$ and $\forall t \in [0, T], |g(t)| > 0$.

    (v) $\forall t \in [0, T] : \boldsymbol{s}_\theta(\cdot, t) \in \mathcal{C}^1, \exists C > 0, \forall \boldsymbol{x} \in \mathbb{R}^n, t \in [0, T] : ||\boldsymbol{s}_\theta(\boldsymbol{x}, t)||_2 \leqslant C(1 + ||\boldsymbol{x}||_2)$.

    (vi) $\exists C > 0, \forall \boldsymbol{x}, \boldsymbol{y} \in \mathbb{R}^n : ||\boldsymbol{s}_\theta(\boldsymbol{x}, t) - \boldsymbol{s}_\theta(\boldsymbol{y}, t)||_2 \leqslant C||\boldsymbol{x} - \boldsymbol{y}||_2$.

## C.2  Background

**Discrete-Time Diffusion Model (or DDPM (Ho et al., 2020)).** DDPM constructs a discrete Markov chain $\{\mathbf{x}_0, \mathbf{x}_1, \cdots, \mathbf{x}_i, \cdots, \mathbf{x}_N\}$ as the forward process for the training data $\mathbf{x}_0 \sim p$, such that $\mathbb{P}(\mathbf{x}_i | \mathbf{x}_{i-1}) = \mathcal{N}(\mathbf{x}_i; \sqrt{1 - \beta_i}\mathbf{x}_{i-1}, \beta_i I)$, where $0 < \beta_1 < \beta_2 < \cdots < \beta_N < 1$ are predefined noise scales such that $\mathbf{x}_N$ approximates the Gaussian white noise. Denote $\overline{\alpha}_i = \prod_{i=1}^N (1 - \beta_i)$, we have $\mathbb{P}(\mathbf{x}_i | \mathbf{x}_0) = \mathcal{N}(\mathbf{x}_i; \sqrt{\overline{\alpha}_i}\mathbf{x}_0, (1 - \overline{\alpha}_i)\boldsymbol{I})$, i.e., $\mathbf{x}_t(\mathbf{x}_0, \epsilon) = \sqrt{\overline{\alpha}_i}\mathbf{x}_0 + (1 - \overline{\alpha}_i)\epsilon, \epsilon \sim \mathcal{N}(\mathbf{0}, \boldsymbol{I})$.

The reverse process of DDPM learns a reverse direction variational Markov chain $p_{\boldsymbol{\theta}}(\mathbf{x}_{i-1} | \mathbf{x}_i) = \mathcal{N}(\mathbf{x}_{i-1}; \boldsymbol{\mu_\theta}(\mathbf{x}_i, i), \Sigma_{\boldsymbol{\theta}}(\mathbf{x}_i, i))$. Ho et al. (2020) defines $\epsilon_{\boldsymbol{\theta}}$ as a function approximator to predict $\epsilon$ from $\boldsymbol{x}_i$ such that $\boldsymbol{\mu_\theta}(\mathbf{x}_i, i) = \frac{1}{\sqrt{1 - \beta_i}}\left(\mathbf{x}_i - \frac{\beta_i}{\sqrt{1 - \overline{\alpha}_i}}\epsilon_{\boldsymbol{\theta}}(\mathbf{x}_i, i)\right)$. Then the reverse time samples are generated by $\hat{\mathbf{x}}_{i-1} = \frac{1}{\sqrt{1 - \beta_i}}\left(\hat{\mathbf{x}}_i - \frac{\beta_i}{\sqrt{1 - \overline{\alpha}_i}}\epsilon_{\boldsymbol{\theta}^*}(\hat{\mathbf{x}}_i, i)\right) + \sqrt{\beta_i}\epsilon, \epsilon \sim \mathcal{N}(\mathbf{0}, I)$, and the optimal parameters $\boldsymbol{\theta}^*$ are obtained by solving $\boldsymbol{\theta}^* := \arg\min_{\boldsymbol{\theta}} \mathbb{E}_{\mathbf{x}_0, \epsilon}\left[||\epsilon - \epsilon_{\boldsymbol{\theta}}(\sqrt{\overline{\alpha}_i}\mathbf{x}_0 + (1 - \overline{\alpha}_i), i)||_2^2\right]$.

**Randomized Smoothing.** Randomized smoothing is used to certify the robustness of a given classifier against $L_2$-norm based perturbation. It transfers the classifier $f$ to a smooth version $g(\boldsymbol{x}) = \arg\max_c \mathbb{P}_{\boldsymbol{\epsilon} \sim \mathcal{N}(\mathbf{0}, \sigma^2 I)}(f(\boldsymbol{x} + \boldsymbol{\epsilon}) = c)$, where $g$ is the smooth classifier and $\sigma$ is a hyperparameter of the smooth classifier $g$, which controls the trade-off between robustness and accuracy. Cohen et al. (2019) shows that $g(x)$ induces the certifiable robustness for $\boldsymbol{x}$ under the $L_2$-norm with radius $R$, where $R = \frac{\sigma}{2}\left(\Phi^{-1}(p_A) - \Phi^{-1}(p_B)\right)$; $p_A$ and $p_B$ are probability of the most probable class and "runner-up" class respectively; $\Phi$ is the inverse of the standard Gaussian CDF. The $p_A$ and $p_B$ can be estimated with arbitrarily high confidence via Monte Carlo method (Cohen et al., 2019).

## C.3 Theorems and Proofs

409 **Theorem 3.1.** *Under conditions C.1, solving equation reverse-SDE starting from time $t$ and point*
410 $\boldsymbol{x}_{a,t} = \sqrt{\alpha_t}\boldsymbol{x}_a$ *will generate a reversed random variable* $\hat{\mathbf{x}}_0$ *with conditional distribution*

$$\mathbb{P}\left(\hat{\mathbf{x}}_0 = \boldsymbol{x}|\hat{\mathbf{x}}_t = \boldsymbol{x}_{a,t}\right) \propto p(\boldsymbol{x}) \cdot \frac{1}{\sqrt{(2\pi\sigma_t^2)^n}} e^{\frac{-||\boldsymbol{x}-\boldsymbol{x}_a||_2^2}{2\sigma_t^2}}$$

411 *where* $\sigma_t^2 = \frac{1-\alpha_t}{\alpha_t}$ *is the variance of the Gaussian noise added at timestamp* $t$ *in the diffusion*
412 *process SDE.*

413 *Proof.* Under the assumption, we know $\{\mathbf{x}_t\}_{t\in[0,1]}$ and $\{\hat{\mathbf{x}}_t\}_{t\in[0,1]}$ follow the same distribution,
414 which means

$$\begin{aligned}
\mathbb{P}\left(\hat{\mathbf{x}}_0 = \boldsymbol{x}|\hat{\mathbf{x}}_t = \boldsymbol{x}_{a,t}\right) &= \frac{\mathbb{P}(\hat{\mathbf{x}}_0 = \boldsymbol{x}, \hat{\mathbf{x}}_t = \boldsymbol{x}_{a,t})}{\mathbb{P}(\hat{\mathbf{x}}_t = \boldsymbol{x}_{a,t})} \\
&= \frac{\mathbb{P}(\mathbf{x}_0 = \boldsymbol{x}, \mathbf{x}_t = \boldsymbol{x}_{a,t})}{\mathbb{P}(\mathbf{x}_t = \boldsymbol{x}_{a,t})} \\
&= \mathbb{P}\left(\mathbf{x}_0 = \boldsymbol{x}\right) \frac{\mathbb{P}(\mathbf{x}_t = \boldsymbol{x}_{a,t}|\mathbf{x}_0 = \boldsymbol{x})}{\mathbb{P}(\mathbf{x}_t = \boldsymbol{x}_{a,t})} \\
&\propto \mathbb{P}\left(\mathbf{x}_0 = \boldsymbol{x}\right) \frac{1}{\sqrt{(2\pi\sigma_t^2)^n}} e^{\frac{-||\boldsymbol{x}-\boldsymbol{x}_a||_2^2}{2\sigma_t^2}} \\
&= p(\boldsymbol{x}) \cdot \frac{1}{\sqrt{(2\pi\sigma_t^2)^n}} e^{\frac{-||\boldsymbol{x}-\boldsymbol{x}_a||_2^2}{2\sigma_t^2}}
\end{aligned}$$

415 where the third equation is due to the chain rule of probability and the last equation is a result of the
416 diffusion process. □

417 **Theorem 3.3.** *Under conditions C.1 and classifier* $f$, *let* $\boldsymbol{x}_0$ *be the sample with ground-truth label*
418 *and* $\boldsymbol{x}_a$ *be the adversarial sample, then (i) the purified sample* $\mathcal{P}(\boldsymbol{x}_a; t)$ *will have the ground-truth*
419 *label if* $\boldsymbol{x}_a$ *falls into the following convex set,*

$$\mathcal{D}_{sub}\left(\boldsymbol{x}_0; t\right) := \bigcap_{\left\{\boldsymbol{x}_0': f(\boldsymbol{x}_0')\neq f(\boldsymbol{x}_0)\right\}} \left\{\boldsymbol{x}_a : (\boldsymbol{x}_a - \boldsymbol{x}_0)^\top(\boldsymbol{x}_0' - \boldsymbol{x}_0) < \sigma_t^2 \log\left(\frac{p(\boldsymbol{x}_0)}{p(\boldsymbol{x}_0')}\right) + \frac{||\boldsymbol{x}_0' - \boldsymbol{x}_0||_2^2}{2}\right\},$$

420 *and further, (ii) the purified sample* $\mathcal{P}(\boldsymbol{x}_a; t)$ *will have the ground-truth label if and only if* $\boldsymbol{x}_a$ *falls*
421 *into the following set,* $\mathcal{D}\left(\mathcal{G}(\boldsymbol{x}_0); t\right) := \bigcup_{\tilde{\boldsymbol{x}}_0: f(\tilde{\boldsymbol{x}}_0)=f(\boldsymbol{x}_0)} \mathcal{D}_{sub}\left(\tilde{\boldsymbol{x}}_0; t\right)$. *In other words,* $\mathcal{D}\left(\mathcal{G}(\boldsymbol{x}_0); t\right)$
422 *is the robust region for data region* $\mathcal{G}(\boldsymbol{x}_0)$ *under* $\mathcal{P}(\cdot; t)$ *and* $f$.

423 *Proof.* We start with part (i).

424 The main idea is to prove that a point $\boldsymbol{x}_0'$ such that $f(\boldsymbol{x}_0') \neq f(\boldsymbol{x}_0)$ should have lower density than
425 $\boldsymbol{x}_0$ in the conditional distribution in Theorem 3.1 so that $\mathcal{P}(\boldsymbol{x}_a; t)$ cannot be $\boldsymbol{x}_0'$. In other words, we
426 should have

$$\mathbb{P}\left(\hat{\mathbf{x}}_0 = \boldsymbol{x}_0|\hat{\mathbf{x}}_t = \boldsymbol{x}_{a,t}\right) > \mathbb{P}\left(\hat{\mathbf{x}}_0 = \boldsymbol{x}_0' \mid \hat{\mathbf{x}}_t = \boldsymbol{x}_{a,t}\right).$$

427 By Theorem 3.1, this is equivalent to

$$\begin{aligned}
& p(\boldsymbol{x}_0) \cdot \frac{1}{\sqrt{(2\pi\sigma_t^2)^n}} e^{\frac{-||\boldsymbol{x}_0-\boldsymbol{x}_a||_2^2}{2\sigma_t^2}} > p(\boldsymbol{x}_0') \cdot \frac{1}{\sqrt{(2\pi\sigma_t^2)^n}} e^{\frac{-||\boldsymbol{x}_0'-\boldsymbol{x}_a||_2^2}{2\sigma_t^2}} \\
\Leftrightarrow & \log\left(\frac{p(\boldsymbol{x}_0)}{p(\boldsymbol{x}_0')}\right) > \frac{1}{2\sigma_t^2}\left(||\boldsymbol{x}_0 - \boldsymbol{x}_a||_2^2 - ||\boldsymbol{x}_0' - \boldsymbol{x}_a||_2^2\right) \\
\Leftrightarrow & \log\left(\frac{p(\boldsymbol{x}_0)}{p(\boldsymbol{x}_0')}\right) > \frac{1}{2\sigma_t^2}\left(||\boldsymbol{x}_0 - \boldsymbol{x}_a||_2^2 - ||\boldsymbol{x}_0' - \boldsymbol{x}_0 + \boldsymbol{x}_0 - \boldsymbol{x}_a||_2^2\right) \\
\Leftrightarrow & \log\left(\frac{p(\boldsymbol{x}_0)}{p(\boldsymbol{x}_0')}\right) > \frac{1}{2\sigma_t^2}\left(2(\boldsymbol{x}_a - \boldsymbol{x}_0)^\top(\boldsymbol{x}_0' - \boldsymbol{x}_0) - ||\boldsymbol{x}_0' - \boldsymbol{x}_0||_2^2\right).
\end{aligned}$$

428   Re-organizing the above inequality, we obtain

$$(\boldsymbol{x}_a - \boldsymbol{x}_0)^\top (\boldsymbol{x}'_0 - \boldsymbol{x}_0) < \sigma_t^2 \log\left(\frac{p(\boldsymbol{x}_0)}{p(\boldsymbol{x}'_0)}\right) + \frac{1}{2}||\boldsymbol{x}'_0 - \boldsymbol{x}_0||_2^2.$$

429   Note that the order of $\boldsymbol{x}_a$ is at most one in every term of the above inequality, so the inequality
430   actually defines a half-space in $\mathbb{R}^n$ for every $(\boldsymbol{x}_0, \boldsymbol{x}'_0)$ pair. Further, we have to satisfy the inequality
431   for every $\boldsymbol{x}'_0$ such that $f(\boldsymbol{x}'_0) \neq f(\boldsymbol{x}_0)$, therefore, by intersecting over all such half-spaces, we
432   obtain a convex $\mathcal{D}_{\text{sub}}(\boldsymbol{x}_0; t)$.

433   Then we prove part (ii).

434   On the one hand, if $\boldsymbol{x}_a \in \mathcal{D}(\mathcal{G}(\boldsymbol{x}_0); t)$, then there exists one $\tilde{\boldsymbol{x}}_0$ such that $f(\tilde{\boldsymbol{x}}_0) = f(\boldsymbol{x}_0)$ and
435   $\boldsymbol{x}_a \in \mathcal{D}_{\text{sub}}(\tilde{\boldsymbol{x}}_0; t)$. By part (i), $\tilde{\boldsymbol{x}}_0$ has higher probability than all other points with different la-
436   bels from $\boldsymbol{x}_0$ in the conditional distribution $\mathbb{P}(\hat{\mathbf{x}}_0 = \boldsymbol{x}|\hat{\mathbf{x}}_t = \boldsymbol{x}_{a,t})$ characterized by Theorem 3.1.
437   Therefore, $\mathcal{P}(\boldsymbol{x}_a; t)$ should have the same label as $\boldsymbol{x}_0$. On the other hand, if $\boldsymbol{x}_a \notin \mathcal{D}(\mathcal{G}(\boldsymbol{x}_0); t)$,
438   then there is a point $\tilde{\boldsymbol{x}}_1$ with different label from $\boldsymbol{x}_0$ such that for any $\tilde{\boldsymbol{x}}_0$ with the same label as $\boldsymbol{x}_0$,
439   $\mathbb{P}(\hat{\mathbf{x}}_0 = \tilde{\boldsymbol{x}}_1|\hat{\mathbf{x}}_t = \boldsymbol{x}_{a,t}) > \mathbb{P}(\hat{\mathbf{x}}_0 = \tilde{\boldsymbol{x}}_0|\hat{\mathbf{x}}_t = \boldsymbol{x}_{a,t})$. In other words, $\mathcal{P}(\boldsymbol{x}_a; t)$ would have different
440   label from $\boldsymbol{x}_0$.   $\square$

441   **Theorem 3.4.** *Under score-based diffusion model Song et al. (2021b) and conditions C.1, we can*
442   *bound*

$$D_{KL}(\mathbb{P}(\hat{\mathbf{x}}_0 = \boldsymbol{x} \mid \hat{\mathbf{x}}_t = \boldsymbol{x}_{a,t})\|\mathbb{P}(\mathbf{x}_0^\theta = \boldsymbol{x} \mid \mathbf{x}_t^\theta = \boldsymbol{x}_{a,t})) = \mathcal{J}_{\text{SM}}(\theta, t; \lambda(\cdot))$$

*where $\{\hat{\boldsymbol{x}}_\tau\}_{\tau \in [0,t]}$ and $\{\boldsymbol{x}_\tau^\theta\}_{\tau \in [0,t]}$ are stochastic processes generated by reverse-SDE and score-*
*based diffusion model respectively,*

$$\mathcal{J}_{\text{SM}}(\theta, t; \lambda(\cdot)) := \frac{1}{2}\int_0^t \mathbb{E}_{p_\tau(\mathbf{x})}\left[\lambda(\tau)\|\nabla_{\mathbf{x}}\log p_\tau(\mathbf{x}) - \boldsymbol{s}_\theta(\mathbf{x}, \tau)\|_2^2\right]\mathrm{d}\tau,$$

443   *$\boldsymbol{s}_\theta(\mathbf{x}, \tau)$ is the score function to approximate $\nabla_{\mathbf{x}}\log p_\tau(\mathbf{x})$, and $\lambda : \mathbb{R} \to \mathbb{R}$ is any weighting scheme*
444   *used in the training score-based diffusion models.*

445   *Proof.* Similar to proof of (Song et al., 2021a, Theorem 1), let $\boldsymbol{\mu}_t$ and $\boldsymbol{\nu}_t$ be the path measure for
446   reverse processes $\{\hat{\mathbf{x}}_\tau\}_{\tau \in [0,t]}$ and $\{\mathbf{x}_\tau^\theta\}_{\tau \in [0,t]}$ respectively based on the scaled adversarial sample
447   $\boldsymbol{x}_{a,t}$. Under conditions C.1, the KL-divergence can be computed via the Girsanov theorem Oksendal
448   (2013):

$$D_{\text{KL}}\left(\mathbb{P}(\hat{\mathbf{x}}_0 = \boldsymbol{x} \mid \hat{\mathbf{x}}_t = \boldsymbol{x}_{a,t})\|\mathbb{P}(\mathbf{x}_0^\theta = \boldsymbol{x} \mid \mathbf{x}_t^\theta = \boldsymbol{x}_{a,t})\right)$$

$$= -\mathbb{E}_{\boldsymbol{\mu}_t}\left[\log\frac{d\boldsymbol{\nu}_t}{d\boldsymbol{\mu}_t}\right]$$

$$\overset{(i)}{=} \mathbb{E}_{\boldsymbol{\mu}_t}\left[\int_0^t g(\tau)(\nabla_{\mathbf{x}}\log p_\tau(\mathbf{x}) - \boldsymbol{s}_\theta(\mathbf{x}, \tau))\,\mathrm{d}\overline{\mathbf{w}}_\tau + \frac{1}{2}\int_0^t g(\tau)^2\|\nabla_{\mathbf{x}}\log p_\tau(\mathbf{x}) - \boldsymbol{s}_\theta(\mathbf{x}, \tau)\|_2^2\,\mathrm{d}\tau\right]$$

$$= \mathbb{E}_{\boldsymbol{\mu}_t}\left[\frac{1}{2}\int_0^t g(\tau)^2\|\nabla_{\mathbf{x}}\log p_\tau(\mathbf{x}) - s_\theta(\mathbf{x}, \tau)\|_2^2\,\mathrm{d}\tau\right]$$

$$= \frac{1}{2}\int_0^\tau \mathbb{E}_{p_\tau(\mathbf{x})}\left[g(\tau)^2\|\nabla_{\mathbf{x}}\log p_\tau(\mathbf{x}) - s_\theta(\mathbf{x}, \tau)\|_2^2\right]\mathrm{d}\tau$$

$$= \mathcal{J}_{\text{SM}}\left(\theta, t; g(\cdot)^2\right)$$

449   where (i) is due to Girsanov Theorem and (ii) is due to the martingale property of Itô integrals.   $\square$

# D   More details about **DensePure**

## D.1   Pseudo-Code

452   We provide the pseudo code of DensePure in Algo. 1 and Alg. 2

**Algorithm 1** DensePure pseudo-code with the highest density point

1: Initialization: choose off-the-shelf diffusion model and classifier $f$, choose $\psi = t$,
2: Input sample $\boldsymbol{x}_a = \boldsymbol{x}_0 + \boldsymbol{\epsilon}_a$
3: Compute $\hat{\boldsymbol{x}}_0 = \mathcal{P}(\boldsymbol{x}_a; \psi)$
4: $\hat{y} = f(\hat{\boldsymbol{x}}_0)$

---

**Algorithm 2** DensePure pseudo-code with majority vote

1: Initialization: choose off-the-shelf diffusion model and classifier $f$, choose $\sigma$
2: Compute $\overline{\alpha}_n = \frac{1}{1+\sigma^2}$, $n = \arg\min_s \left\{ \left| \overline{\alpha}_s - \frac{1}{1+\sigma^2} \right| \mid s \in \{1, 2, \cdots, N\} \right\}$
3: Generate input sample $\boldsymbol{x}_{rs} = \boldsymbol{x}_0 + \boldsymbol{\epsilon}, \boldsymbol{\epsilon} \sim \mathcal{N}(\mathbf{0}, \sigma^2\boldsymbol{I})$
4: Choose schedule $S^b$, get $\hat{\boldsymbol{x}}_0^i \leftarrow \mathbf{rev}(\sqrt{\overline{\alpha}_n}\boldsymbol{x}_{rs})_i, i = 1, 2, \ldots, K$ with Fast Sampling
5: $\hat{y} = \mathbf{MV}(\{f(\hat{\boldsymbol{x}}_0^1), \ldots, f(\hat{\boldsymbol{x}}_0^K)\}) = \arg\max_c \sum_{i=1}^K \mathbf{1}\{f(\hat{\boldsymbol{x}}_0^i) = c\}$

---

### D.2 Certified Robustness of **DensePure** with Randomized Smoothing.

We show how DensePure can calculate certified robustness of DensePure via RS, which offers robustness guarantees for a model under a $L_2$-norm ball.

In particular, we follow the similar setting of Carlini et al. (2022) which uses a DDPM-based diffusion model. The details are in the appendix. The overall algorithm contains three steps:

(1) Our framework estimates $n$, the number of steps used for the reverse process of DDPM-based diffusion model. Since Randomized Smoothing (Cohen et al., 2019) adds Gaussian noise $\boldsymbol{\epsilon}$, where $\boldsymbol{\epsilon} \sim \mathcal{N}(\mathbf{0}, \sigma^2\boldsymbol{I})$, to data input $\boldsymbol{x}$ to get the randomized data input, $\boldsymbol{x}_{rs} = \boldsymbol{x} + \boldsymbol{\epsilon}$, we map between the noise required by the randomized example $\boldsymbol{x}_{rs}$ and the noise required by the diffused data $\boldsymbol{x}_n$ (i.e., $\boldsymbol{x}_n \sim \mathcal{N}(\boldsymbol{x}_n; \sqrt{\overline{\alpha}_n}\boldsymbol{x}_0, (1 - \overline{\alpha}_n)\boldsymbol{I})$) with $n$ step diffusion processing so that $\overline{\alpha}_n = \frac{1}{1+\sigma^2}$. In this way, we can compute the corresponding timestep $n$, where $n = \arg\min_s\{|\overline{\alpha}_s - \frac{1}{1+\sigma^2}| \mid s \in \{1, 2, \cdots, N\}\}$.

(2). Given the above calculated timestep $n$, we scale $\boldsymbol{x}_{rs}$ with $\sqrt{\overline{\alpha}_n}$ to obtain the scaled randomized smoothing sample $\sqrt{\overline{\alpha}_n}\boldsymbol{x}_{rs}$. Then we feed $\sqrt{\overline{\alpha}_n}\boldsymbol{x}_{rs}$ into the reverse process of the diffusion model by $K$-times to get the reversed sample set $\{\hat{\boldsymbol{x}}_0^1, \hat{\boldsymbol{x}}_0^2, \cdots, \hat{\boldsymbol{x}}_0^i, \cdots, \hat{\boldsymbol{x}}_0^K\}$.

(3). We feed the obtained reversed sample set into a standard *off-the-shelf* classifier $f$ to get the corresponding predicted labels $\{f(\hat{\boldsymbol{x}}_0^1), f(\hat{\boldsymbol{x}}_0^2), \ldots, f(\hat{\boldsymbol{x}}_0^i), \ldots, f(\hat{\boldsymbol{x}}_0^K)\}$, and apply *majority vote*, denoted $\mathbf{MV}(\cdots)$, on these predicted labels to get the final label for $\boldsymbol{x}_{rs}$.

To calculate the reversed sample, the standard reverse process of DDPM-based models require repeatedly applying a "single-step" operation $n$ times to get the reversed sample $\hat{\boldsymbol{x}}_0$ (i.e., $\hat{\boldsymbol{x}}_0 = \underbrace{\mathbf{Reverse}(\cdots \mathbf{Reverse}(\cdots \mathbf{Reverse}(\mathbf{Reverse}(\sqrt{\overline{\alpha}_n}\boldsymbol{x}_{rs}; n); n-1); \cdots; i); \cdots 1)}_{n \text{ steps}}$). Here $\hat{\boldsymbol{x}}_{i-1} = \mathbf{Reverse}(\hat{\boldsymbol{x}}_i; i)$ is equivalent to sample $\hat{\boldsymbol{x}}_{i-1}$ from $\mathcal{N}(\hat{\boldsymbol{x}}_{i-1}; \boldsymbol{\mu_\theta}(\hat{\boldsymbol{x}}_i, i), \boldsymbol{\Sigma_\theta}(\hat{\boldsymbol{x}}_i, i))$, where $\boldsymbol{\mu_\theta}(\hat{\boldsymbol{x}}_i, i) = \frac{1}{\sqrt{1-\beta_i}}\left(\hat{\boldsymbol{x}}_i - \frac{\beta_i}{\sqrt{1-\overline{\alpha}_i}}\boldsymbol{\epsilon_\theta}(\hat{\boldsymbol{x}}_i, i)\right)$ and $\boldsymbol{\Sigma_\theta} := \exp(v \log \beta_i + (1-v) \log \widetilde{\beta}_i)$. Here $v$ is a parameter learned by DDPM and $\widetilde{\beta}_i = \frac{1-\overline{\alpha}_{i-1}}{1-\overline{\alpha}_i}$.

To reduce the time complexity, we use the uniform sub-sampling strategy from Nichol & Dhariwal (2021). We uniformly sample a subsequence with size $b$ from the original $N$-step the reverse process. In details, we follow the method used in (Nichol & Dhariwal, 2021) and sample a subsequence $S^b$ with $b$ values (i.e., $S^b = \underbrace{\{n, \lfloor n - \frac{n}{b} \rfloor, \cdots, 1\}}_{b}$, where $S_i^b$ is the $i$-th element in $S^b$ and $S_i^b = \lfloor n - \frac{in}{b} \rfloor, \forall i < b$ and $S_b^b = 1$) from the original schedule $S$ (i.e., $S = \underbrace{\{n, n-1, \cdots, 1\}}_{n}$, where $S_i = i$ is the $i$-th element in $S$).

| Methods | Noise | Certified Accuracy at $\epsilon$(%) | | | | |
| --- | --- | --- | --- | --- | --- | --- |
| | | 0.0 | 0.25 | 0.5 | 0.75 | 1.0 |
| Carlini (Carlini et al., 2022) | $\sigma = 0.25$ | **88.0** | 73.8 | 56.2 | 41.6 | 0.0 |
| | $\sigma = 0.5$ | 74.2 | 62.0 | 50.4 | 40.2 | 31.0 |
| | $\sigma = 1.0$ | 49.4 | 41.4 | 34.2 | 27.8 | 21.8 |
| **Ours** | $\sigma = 0.25$ | 87.6(-0.4) | **76.6(+2.8)** | **64.6(+8.4)** | **50.4(+8.8)** | 0.0(+0.0) |
| | $\sigma = 0.5$ | 73.6(-0.6) | 65.4(+3.4) | 55.6(+5.2) | 46.0(+5.8) | **37.4(+6.4)** |
| | $\sigma = 1.0$ | 55.0(+5.6) | 47.8(+6.4) | 40.8(+6.6) | 33.0(+5.2) | 28.2(+6.4) |

Table A: Certified accuracy compared with Carlini et al. (2022) for CIFAR-10 at all $\sigma$. The numbers in the bracket are the difference of certified accuracy between two methods. Our diffusion model and classifier are the same as Carlini et al. (2022).

Within this context, we adapt the original $\overline{\alpha}$ schedule $\overline{\alpha}^S = \{\overline{\alpha}_1, \cdots, \overline{\alpha}_i, \cdots, \overline{\alpha}_n\}$ used for single-step to the new schedule $\overline{\alpha}^{S^b} = \{\overline{\alpha}_{S_1^b}, \cdots, \overline{\alpha}_{S_j^b}, \cdots, \overline{\alpha}_{S_b^b}\}$ (i.e., $\overline{\alpha}_i^{S^b} = \overline{\alpha}_{S_i^b} = \overline{\alpha}_{S_{\lfloor n - \frac{in}{b} \rfloor}}$ is the $i$-th element in $\overline{\alpha}^{S^b}$). We calculate the corresponding $\beta^{S^b} = \{\beta_1^{S^b}, \beta_2^{S^b}, \cdots, \beta_i^{S^b}, \cdots, \beta_b^{S^b}\}$ and $\widetilde{\beta}^{S^b} = \{\widetilde{\beta}_1^{S^b}, \widetilde{\beta}_2^{S^b}, \cdots, \widetilde{\beta}_i^{S^b}, \cdots, \widetilde{\beta}_b^{S^b}\}$ schedules, where $\beta_{S_i^b} = \beta_i^{S^b} = 1 - \frac{\overline{\alpha}_i^{S^b}}{\overline{\alpha}_{i-1}^{S^b}}$, $\widetilde{\beta}_{S_i^b} = \widetilde{\beta}_i^{S^b} = \frac{1 - \overline{\alpha}_{i-1}^{S^b}}{1 - \overline{\alpha}_i^{S^b}} \beta_{S_i^b}$. With these new schedules, we can use $b$ times reverse steps to calculate $\hat{x}_0 = \underbrace{\textbf{Reverse}(\cdots \textbf{Reverse}(\textbf{Reverse}(\boldsymbol{x}_n; S_b^b); S_{b-1}^b); \cdots; 1)}_{b}$. Since $\boldsymbol{\Sigma}_{\boldsymbol{\theta}}(\boldsymbol{x}_{S_i^b}, S_i^b)$ is parameterized as a range between $\beta^{S^b}$ and $\widetilde{\beta}^{S^b}$, it will automatically be rescaled. Thus, $\hat{\boldsymbol{x}}_{S_{i-1}^b} = \textbf{Reverse}(\hat{\boldsymbol{x}}_{S_i^b}; S_i^b)$ is equivalent to sample $\boldsymbol{x}_{S_{i-1}^b}$ from $\mathcal{N}(\boldsymbol{x}_{S_{i-1}^b}; \boldsymbol{\mu}_{\boldsymbol{\theta}}(\boldsymbol{x}_{S_i^b}, S_i^b), \boldsymbol{\Sigma}_{\boldsymbol{\theta}}(\boldsymbol{x}_{S_i^b}, S_i^b))$.

# E More Experimental details and Results

## E.1 Implementation details

We select three different noise levels $\sigma \in \{0.25, 0.5, 1.0\}$ for certification. For the parameters of DensePure , The sampling numbers when computing the certified radius are $n = 100000$ for CIFAR-10 and $n = 10000$ for ImageNet. We evaluate the certified robustness on 500 samples subset of CIFAR-10 testset and 100 samples subset of ImageNet validation set. we set $K = 40$ and $b = 10$ except the results in ablation study. The details about the baselines are in the appendix.

## E.2 Baselines.

We select randomized smoothing based methods including PixelDP (Lecuyer et al., 2019), RS (Cohen et al., 2019), SmoothAdv (Salman et al., 2019a), Consistency (Jeong & Shin, 2020), MACER (Zhai et al., 2020), Boosting (Horváth et al., 2021) , SmoothMix (Jeong et al., 2021), Denoised (Salman et al., 2020), Lee (Lee, 2021), Carlini (Carlini et al., 2022) as our baselines. Among them, PixelDP, RS, SmoothAdv, Consistency, MACER, and SmoothMix require training a smooth classifier for a better certification performance while the others do not. Salman et al. and Lee use the off-the-shelf classifier but without using the diffusion model. The most similar one compared with us is Carlini et al., which also uses both the off-the-shelf diffusion model and classifier. The above two settings mainly refer to Carlini et al. (2022), which makes us easier to compared with their results.

## E.3 Main Results for Certified Accuracy

We compare with Carlini et al. (2022) in a more fine-grained version. We provide results of certified accuracy at different $\epsilon$ in Table A for CIFAR-10 and Table B for ImageNet. We include the accuracy difference between ours and Carlini et al. (2022) in the bracket in Tables. We can observe from the tables that the certified accuracy of our method outperforms Carlini et al. (2022) except $\epsilon = 0$ at $\sigma = 0.25, 0.5$ for CIFAR-10.

| Methods | Noise | Certified Accuracy at $\epsilon$(%) | | | | | |
|---|---|---|---|---|---|---|---|
| | | 0.0 | 0.5 | 1.0 | 1.5 | 2.0 | 3.0 |
| Carlini (Carlini et al., 2022) | $\sigma = 0.25$ | 77.0 | 71.0 | 0.0 | 0.0 | 0.0 | 0.0 |
| | $\sigma = 0.5$ | 74.0 | 67.0 | 54.0 | 46.0 | 0.0 | 0.0 |
| | $\sigma = 1.0$ | 59.0 | 53.0 | 49.0 | 38.0 | 29.0 | 22.0 |
| **Ours** | $\sigma = 0.25$ | **80.0(+3.0)** | **76.0(+5.0)** | 0.0(+0.0) | 0.0(+0.0) | 0.0(+0.0) | 0.0(+0.0) |
| | $\sigma = 0.5$ | 75.0(+1.0) | 72.0(+5.0) | **62.0(+8.0)** | **49.0(+3.0)** | 0.0(+0.0) | 0.0(+0.0) |
| | $\sigma = 1.0$ | 61.0(+2.0) | 57.0(+4.0) | 53.0(+4.0) | **49.0(+11.0)** | **37.0(+8.0)** | **26.0(+4.0)** |

Table B: Certified accuracy compared with Carlini et al. (2022) for ImageNet at all $\sigma$. The numbers in the bracket are the difference of certified accuracy between two methods. Our diffusion model and classifier are the same as Carlini et al. (2022).

| Datasets | Methods | Model | Certified Accuracy at $\epsilon$(%) | | | | | | | | |
|---|---|---|---|---|---|---|---|---|---|---|---|
| | | | 0.0 | 0.25 | 0.5 | 0.75 | Model | 0.0 | 0.25 | 0.5 | 0.75 |
| CIFAR-10 | Carlini (Carlini et al., 2022) | ViT-B/16 | **93.0** | 76.0 | 57.0 | 47.0 | WRN28-10 | 86.0 | 66.0 | 55.0 | 37.0 |
| | **Ours** | ViT-B/16 | 92.0 | **82.0** | **69.0** | **56.0** | WRN28-10 | **90.0** | **77.0** | **63.0** | **50.0** |
| ImageNet | Carlini (Carlini et al., 2022) | BEiT | 77.0 | 76.0 | 71.0 | 60.0 | WRN50-2 | 73.0 | 67.0 | 57.0 | 48.0 |
| | **Ours** | BEiT | **80.0** | **78.0** | **76.0** | **71.0** | WRN50-2 | **81.0** | **72.0** | **66.0** | **61.0** |

Table C: Certified accuracy of our method among different classifier. BeiT and ViT are pre-trained on a larger dataset ImageNet-22k and fine-tuned at ImageNet-1k and CIFAR-10 respectively. WideResNet is trained on ImageNet-1k for ImageNet and trained on CIFAR-10 from scratch for CIFAR-10.

### E.4 Ablation study

We conduct ablation study on different Voting samples. **Voting samples ($K$)** We first show how $K$ affects the certified accuracy. For efficiency, we select $b = 10$. We conduct experiments for both datasets. We show the certified accuracy among different $r$ at $\sigma = 0.25$ in Figure H. The results for $\sigma = 0.5, 1.0$ and CIFAR-10 are shown in the Appendix E.5. Comparing with the baseline (Carlini et al., 2022), we find that a larger majority vote number leads to a better certified accuracy. It verifies that DensePure indeed benefits the adversarial robustness and making a good approximation of the label with high density region requires a large number of voting samples. We find that our certified accuracy will almost converge at $r = 40$. Thus, we set $r = 40$ for our experiments. The results with other $\sigma$ show the similar tendency.

**Fast sampling steps ($b$)** To investigate the role of $b$, we conduct additional experiments with $b \in \{2, 5\}$ at $\sigma = 0.25$. The results on ImageNet are shown in Figure H and results for $\sigma = 0.5, 1.0$ and CIFAR-10 are shown in the Appendix E.6. By observing results *with* majority vote, we find that a larger $b$ can lead to a better certified accuracy since a larger $b$ generates images with higher quality. By observing results *without* majority vote, the results show opposite conclusions where a larger $b$ leads to a lower certified accuracy, which contradicts to our intuition. We guess the potential reason is that though more sampling steps can normally lead to better image recovery quality, it also brings more randomness, increasing the probability that the reversed image locates into a data region with the wrong label. These results further verify that majority vote is necessary for a better performance.

**Different architectures** One advantage of DensePure is to use the off-the-shelf classifier so that it can plug in any classifier. We choose Convolutional neural network (CNN)-based architectures: Wide-ResNet28-10 (Zagoruyko & Komodakis, 2016) for CIFAR-10 with $95.1\%$ accuracy and Wide-ResNet50-2 for ImageNet with $81.5\%$ top-1 accuracy, at $\sigma = 0.25$. The results are shown in Table C and Figure E in Appendix E.7. Results for more model architectures and $\sigma$ of ImageNet are also shown in Appendix E.7. We show that our method can enhance the certified robustness of any given classifier trained on the original data distribution. Noticeably, although the performance of CNN-based classifier is lower than Transformer-based classifier, DensePure with CNN-based model as the classifier can outperform Carlini et al. (2022) with ViT-based model as the classifier (except $\epsilon = 0$ for CIFAR-10).

### E.5 Experiments for Voting Samples

Here we provide more experiments with $\sigma \in \{0.5, 1.0\}$ and $b = 10$ for different voting samples $K$ in Figure A and Figure B. The results for CIFAR-10 is in Figure G. We can draw the same conclusion mentioned in the main context .

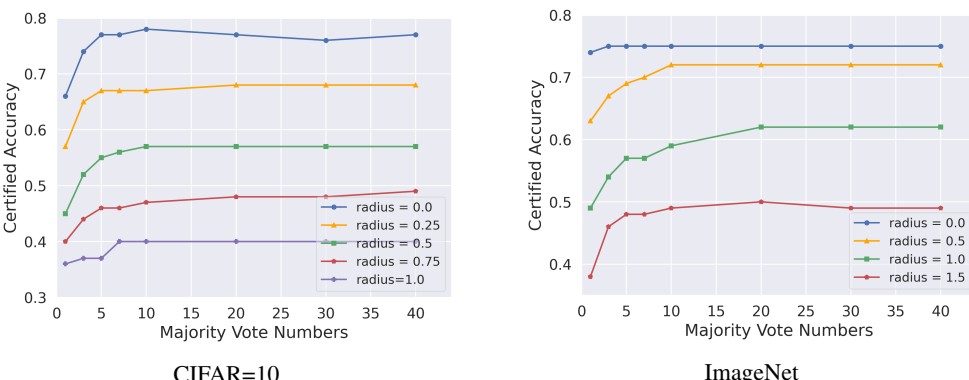

CIFAR=10

ImageNet

Figure A: Certified accuracy among different vote numbers with different radius. Each line in the figure represents the certified accuracy among different vote numbers K with Gaussian noise $\sigma = 0.50$.

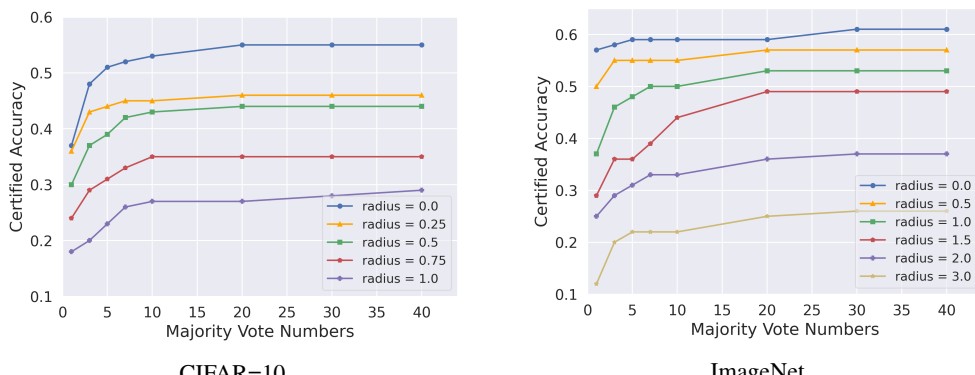

CIFAR=10

ImageNet

Figure B: Certified accuracy among different vote numbers with different radius. Each line in the figure represents the certified accuracy among different vote numbers K with Gaussian noise $\sigma = 1.00$.

### E.6 Experiments for Fast Sampling Steps

We also implement additional experiments with $b \in \{1, 2, 10\}$ at $\sigma = 0.5, 1.0$. The results are shown in Figure C and Figure D. The results for CIFAR-10 are in Figure G. We draw the same conclusion as mentioned in the main context.

### E.7 Experiments for Different Architectures

We try different model architectures of ImageNet including Wide ResNet-50-2 and ResNet 152 with $b = 2$ and $K = 10$. The results are shown in Figure F. we find that our method outperforms (Carlini et al., 2022) for all $\sigma$ among different classifiers.

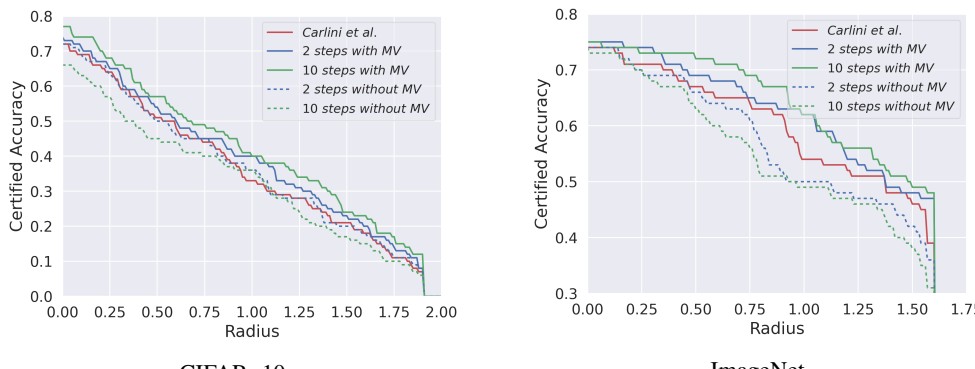

CIFAR=10                    ImageNet

Figure C: Certified accuracy with different fast sampling steps $b$. Each line in the figure shows the certified accuracy among different $L_2$ adversarial perturbation bound with Gaussian noise $\sigma = 0.50$.

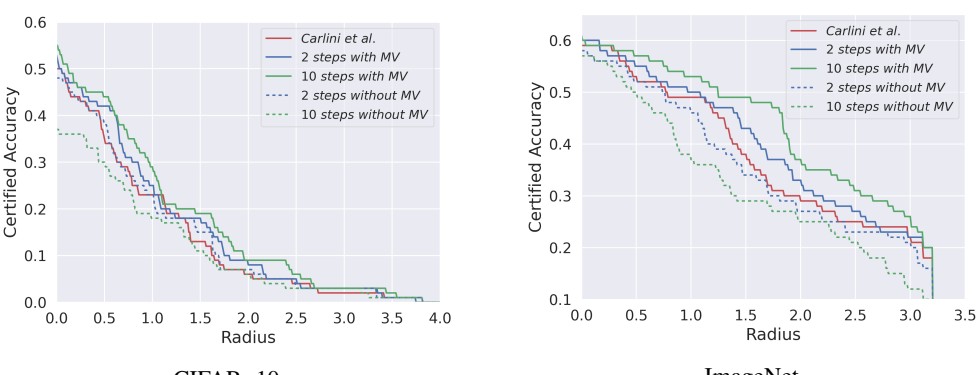

CIFAR=10                    ImageNet

Figure D: Certified accuracy with different fast sampling steps $b$. Each line in the figure shows the certified accuracy among different $L_2$ adversarial perturbation bound with Gaussian noise $\sigma = 1.00$.

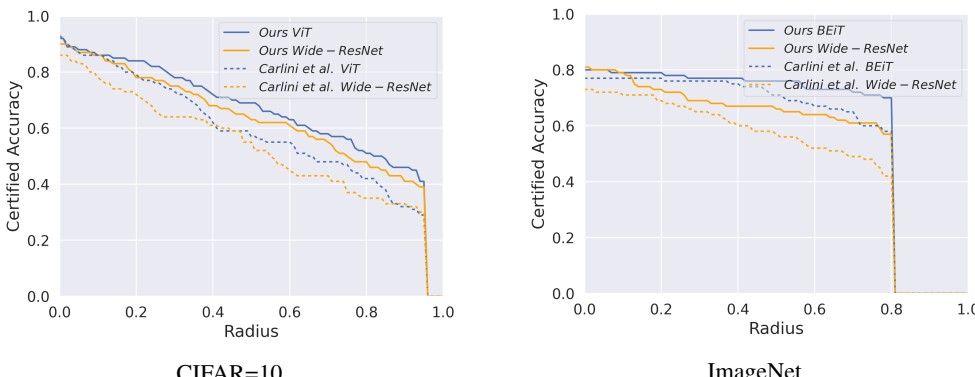

CIFAR=10                    ImageNet

Figure E: Certified accuracy with different architectures. Each line in the figure shows the certified accuracy among different $L_2$ adversarial perturbation bound with Gaussian noise $\sigma = 0.25$.

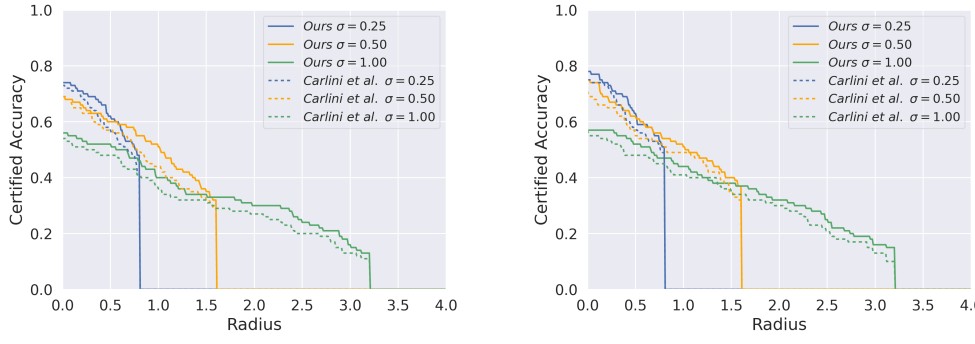

Wide ResNet-50-2            ResNet152

Figure F: Certified accuracy of ImageNet for different architectures. The lines represent the certified accuracy with different $L_2$ perturbation bound with different Gaussian noise $\sigma \in \{0.25, 0.50, 1.00\}$.

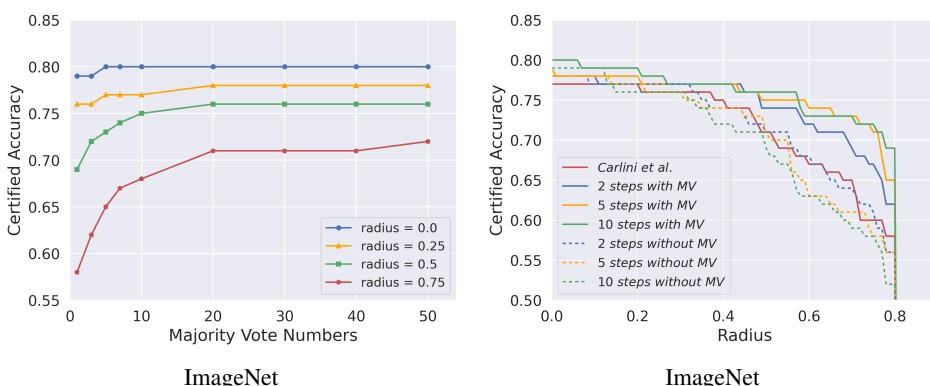

ImageNet            ImageNet

Figure G: Ablation study. The left image shows the certified accuracy among different vote numbers with different radius $\epsilon \in \{0.0, 0.25, 0.5, 0.75\}$. Each line in the figure represents the certified accuracy of our method among different vote numbers $K$ with Gaussian noise $\sigma = 0.25$. The right image shows the certified accuracy with different fast sampling steps $b$. Each line in the figure shows the certified accuracy among different $L_2$ adversarial perturbation bound.

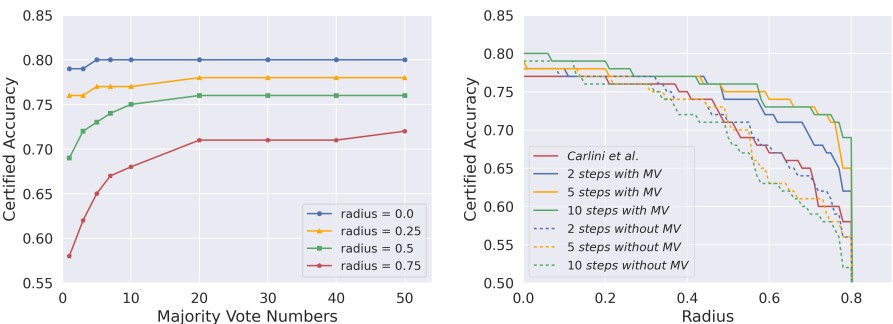

Figure H: Ablation study on ImageNet. The left image shows the certified accuracy among different vote numbers with different radius $\epsilon \in \{0.0, 0.25, 0.5, 0.75\}$. Each line in the figure represents the certified accuracy of our method among different vote numbers $K$ with Gaussian noise $\sigma = 0.25$. The right image shows the certified accuracy with different fast sampling steps $b$. Each line in the figure shows the certified accuracy among different $L_2$ adversarial perturbation bound.

