# OpenReview forum: "DensePure: Understanding Diffusion Models towards Adversarial Robustness "
_NeurIPS.cc/2022/Workshop/TSRML — TSRML2022_

### Official Review · Reviewer_Ttbj · 2022-10-20
**Good Theoretical Analysis of Diffusion Model**

**Overall Recommendation:** Good paper with adequate theoretical …
**Overall Rating:** 7

**Summary:**

This paper analyzes the fundamental properties of the diffusion model used for Certified Defense. Furthermore, the authors utilize this deeper understanding of the diffusion model to propose a method named DensePure for certified defense using the diffusion model.

**Strengths:**

Good Theoretical Analysis of Diffusion Model.

**Weaknesses:**

No figures to demonstrate the pipeline of the proposed method. In addition, the authors need to further clarify the difference between the pipelines of "Carlini et al., 2022" and the proposed method.

**Review Confidence:**

4: The reviewer is confident but not absolutely certain that the evaluation is correct

---

### Official Review · Reviewer_b1B6 · 2022-10-21
**Relevant paper with state-of-the-art performance**

**Overall Recommendation:** The topic is a good match to the work…
**Overall Rating:** 9

**Summary:**

This paper proposed using denoising step of the diffusion models in conjunction with randomized smoothing for obtaining robust classification. The authors provide provable certification, and the reported performance is improving state-of-the-art numbers.

**Strengths:**

The paper is well written, the idea is intuitive, and the claims are backed with theoretical guarantees. Literature review is done well, and expeiment section compares with SOTA and provide improvements.

**Weaknesses:**

Notation is a bit cumbersome

**Review Confidence:**

3: The reviewer is fairly confident that the evaluation is correct

---

### Official Review · Reviewer_Bejr · 2022-10-22
**The paper discusses how diffusion models can be used to improve the certified robustness of models.**

**Overall Rating:** 6

**Summary:**

The paper builds on recent work on determining the certified robustness of
models using  difussion models. In particular the paper provides a theoretical
analysis of how difussion models can be used to derive larger robustness
radiuses than related methods. On the basis of this the paper presents a simple
method that relies on several denoising steps applied on an  adversarial
example  to obtain its robust region.

**Strengths:**

Well written and presented. Even though I am not an expert in the are I could
easily follow the main concepts of the paper.

Good experimental analysis, showing that the proposed method overcomes the SoA
in terms of the robust accuracy that can be established.

**Weaknesses:**

Highly incremental to Carlini et al. 2022.

**Overall Recommendation:**

This is a solid, well-written contribution for the computation of preciser
robust accuracies of models. While in my opinion is highly incremental to
Carlini et al. 2022, this is an opinion with low confidence as I am not
familiar with related work.

**Review Confidence:**

2: The reviewer is willing to defend the evaluation, but it is quite likely that the reviewer did not understand central parts of the paper

---

### Decision · Program_Chairs · 2022-10-23

**Decision:**

Accept

**Comment:**

Great work on enhancing certified defense with state-of-the-art diffusion based generative models.